# Commentary on Sequestering Atmospheric CO₂ Inorganically: A Solution for Malaysia's CO₂ Emission

**John Barry Gallagher \*** , **Nithiyaa Nilamani and Norlaila Binti Mohd Zanuri**

Centre for Marine and Coastal Studies, Universiti Sains, 11800 Gelugor, Malaysia; n.nithiyaa@usm.my (N.N.); lailazanuri@usm.my (N.B.M.Z.)

**\*** Correspondence: john.barry@usm.my

**Abstract:** The commentary questions the basis behind an article on accounting and calculating inorganic carbon sequestration services for Malaysia. We point out the omission of coastal vegetated ecosystems. We also bring the author's attention to the problems of using a seemingly resultant chemistry within open systems, in which reactive species come from external sources. In addition, we point out that ecosystem services in the mitigation of climate change must be referenced against a manufacturing process, such as cement's normal lifetime of carbon dioxide sequestration. Without such a reference state, sequestration services may be severely overestimated and when used within a cap and trade system, it will lead to an increased rate of carbon dioxide emissions.

**Keywords:** inorganic carbon sequestration; blue carbon; sequestration services

## 1. Introduction

Jorat et al. [1] sought to review the potential for Malaysian carbon capture, in particular, the limitations and costs of both industrial and forest ecosystems to current anthropogenic emissions and their sources. They then proposed a relatively inexpensive and more certain solution to make up a significant part of the shortfall through inorganic carbon sequestration. The authors conclude, that the application of solid industrial waste, such as concrete, gypsum, basalt and fly ash soil wastes added to soils could capture nearly 600,000 tonnes annually. The basis of the chemistry comes from Berner et al. [2] suggested modeling of how atmospheric $CO_2$ could have been regulated over geological time scales. They used both by silicate and calcium chemistries separated in geological time and space. In doing so, essentially an equilibrium approach was used based on the amounts of silicates and carbonates formed over a geological cycle of formation of carbonates and silicate weathering (Equation (1)),

$$CO_2 + CaSiO_3 = CaCO_3 + SiO_2. \tag{1}$$

## 2. The Critique

We appreciate the extensive review that also brings to light policies designed to address the issues. However, we feel, first, that the authors have not considered all important natural possible sinks, such as coastal vegetated ecosystems, namely, mangroves, seagrasses and seaweeds. This would be a useful addition and give a more accurate context on which to evaluate any shortfalls towards carbon neutrality across Malaysia. More importantly, we believe that the sequestration values of inorganic wastes that the authors cite are greatly overestimated. This is because of misconceptions of the concept 'ecosystem services in the mitigation of climate change', and issues on the application of this chemistry to open systems. Based on this chemistry, the authors, and others [3], have suggested bringing together

waste materials from the building and energy production industry, ostensibly concretes, cement, and fly ash. This combination, through natural weathering then provides those elements required for the above type of carbonation (equation (1)). Much of the support for this comes ostensibly from the author's previous work. The work, however, appears not to be based on direct measurements of $CO_2$ uptake, but the products in the mixture as inferred by the chemistry, at theoretical equilibrium. However, direct measures under accelerated conditions for similar materials (CSA Type 10 cement, CSA Type 30 cement, fly ash, ground granulated blast furnace (GGBF) slag, electric arc furnace (EAF) slag) have not achieved anything like theoretical maximums [3]. Furthermore, in open soil systems, it is not clear or certain what would be the dominant ionic species, or their intrinsic chemical activities. For example, should the supply of bicarbonate come from ground waters, another element suggested by the authors, then this may well dominate the calcification reaction. This could lead to a reduction or an increase in $CO_2$ emissions (Equation (2)) [4],

$$Ca^{2+} + 2HCO_3^- = CaCO_3 + H_2O + CO_2. \qquad (2)$$

More importantly, the concept of a sequestration service in the mitigation of climate change is based on the differences in sequestration by the alternative ecosystem or state sequestered or emitted over climatic scales [5,6]. For wastes, from processes and concrete structures, which have already been produced, the sequestration mitigation service then becomes the difference between the uptake of $CO_2$, over the centennial lifetime of the product or structure [7]. As this happens in any event, additions of these wastes to soils would not result in a net gain or a least a substantially reduced gain in the sequestration services, be it to some unknown extent.

We acknowledge the difficulty in estimating sequestration from complex soil chemistries. Nevertheless, we believe that the author's sequestration estimates are both too uncertain and suffer from severe overreach. As such overestimates that appear, at first, to be well argued can have aberrant consequences for national policy decisions. For when linked to any future cap and trade system, overestimates of the method's sequestration service will allow companies in the pursuit of carbon neutrality, to increase the nation's overall $CO_2$ emissions.

**Author Contributions:** Conceptualization and writing-original draft preparation, J.B.G. Contributions to the conceptualization of the chemistry and manuscript, N.N.; with contributions to conceptualization and editing, N.B.N.Z.

**Funding:** This research received no funding.

**Conflicts of Interest:** The authors declare no conflict of interest.

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
