# Peer review of "Commentary on Sequestering Atmospheric CO2 Inorganically: A Solution for Malaysia’s CO2 Emission"

_geosciences, doi:10.3390/geosciences9020090_

Round 1
Reviewer 1 Report
The authors present an interesting commentary on the article of Jorat et al. (2018). They mention that the CO2 sequestration values given in the aforementioned paper are overestimated, mainly due to misconceptions of ecosystem service in the mitigation of climate change. I recommend the acceptance of this commentary following some modification from its current form.
List of comments:
- Equation (1): Why do the authors give this specific chemical reaction, which describes the carbonation of wollastonite? They could potentially give the carbonation reactions of other minerals (e.g., anorthite, clinopyroxene) that can be found in some of the waste materials (i.e., basalt quarry fines) described by Jorat et al. (2018).
- P2, L45: The authors should give in parentheses the elements required for the mineralization of CO2. Also, it would be useful for the reader to add the name of the minerals involved in the relevant reactions.
-P2, L47: The reference related to this work should be added.
- Also, the authors should improve the English of their commentary and check the text carefully for grammatical errors.
Author Response
Reply to Reviewer 1
We thank the reviewer for the comments required to improve the manuscript. The specific chemical reaction in equation 1) was to illustrate the net uptake of carbon dioxide based on a theory originally proposed by Berner that models carbon dioxide variability from the net affects of mantle formation sequent uplift and weathering. As this was not a concern within the Reply from the Authors, we beg the Reviewers indulgence to let this principle illustration of the net stoichiometry remain as equation 1.
We added the list of materials in parenthesis, as suggested by the reviewer, as quoted in the reference regarding the lack of attainment of equilibrium
We added the reference for equation 2 as well regarded concern for the calcareous marine ecosystems. These are typically calcites or aragonites. With respect to the reviewers concerns, a general sense doesnt add anything by trying to guess expected mineral would have been for soils, and doesn’t not appear to be a concern of the authors in their reply,
We apologise for the tardy state of the grammar. We have endeavoured to correct any mistakes first through “Grammarly’ and then reading it through to remove or add the occasional addition of words or phrase within sentences.
Reviewer 2 Report
This is a commentary related to a paper recently publihed in Geosciences.
The current authors are from Malaysia, so their comments could be considered correct for the specific case.
Of course, it is difficult to estimate sequestration from complex soil chemistries ...
Author Response
Reply to Reviwer 2
We thank the reviewer for his service and are gratified by his decision to allow publication
We apologise for the tardy state of the grammar and spelling. We have endeavoured to correct any mistakes first through “Grammarly’ and then reading it through to remove or add the occasional addition of words or phrase within sentences.
Reviewer 3 Report
I think the authors present good (and different) vantage points in this work that supplements the previously published work.
Author Response
Reply to Reviewer 3
We thank the reviewer for areas required to improve the manuscript.
We have added another citation with regards the calcium carbonate formation from bicarbonate, added more context with regards to the mix of materials similar to that used in the original article, and tidied up the grammar and readability throughout.
Reviewer 4 Report
Please find the detailed response attached.

Author Response
We thank the authors for their consideration of our critique for this important topic.
To clarify some minor points:
We apologise for the inclusion of quicklime, and take the authors point on the matter, and we have removed it from the article accordingly.
We take the point made by the authors of their concerns of not overreaching with their carefully crafted title and statements. Perhaps too subtle, and when put together with values in dollars terms, and context to natural sequestration, the subtlety may be masked from the reader, somewhat by the rhetoric.
Application to coastal vegetated ecosystems: We did not suggest that the authors should attempt to bury these materials within mangrove and seagrass sediments. The idea was to bring the authors attention to these highly productive and significant sequester, and importantly storage systems. Often terrestrial and marine disciplines fail to connect. This can lead to incorrect assessments and proper context. The two references we cited to these blue carbon ecosystems should have sufficient information and citations for the authors to further research these environments, along with some of the issues within the current state of the science
We stand by our main point that net sequestration services need to be calculated from not only a difference in the baseline, but the difference in what would have happen in any event from not burying these wastes. At equilibrium, they should be no net benefit. But as indicated (and in citations) in any sequestration service in the mitigation of climate change, needs unit timescale (centennial from IPCC), and the equilibrium state at the end of that timescale.
We are unclear by the authors reply ‘that direct measurements of carbon dioxide may have been made in some cases’. We would be grateful, at a later date, for the authors to cite these examples to our institutional email addresses, and perhaps start a dialogue on the subject both terrestrial and marine aspects, as indicated by the authors.